# Statistical Knowledge Assessment
# for Large Language Models

**Qingxiu Dong**[1]**, Jingjing Xu**[2]**, Lingpeng Kong**[3]**, Zhifang Sui**[1] **and Lei Li**[4]

[1] National Key Laboratory for Multimedia Information Processing,
School of Computer Science, Peking University
[2] Shanghai AI Lab   [3] The University of Hong Kong   [4] Carnegie Mellon University
dqx@stu.pku.edu.cn, {jingjingxu, szf}@pku.edu.cn, lpk@cs.hku.hk, leili@cs.cmu.edu

## Abstract

Given varying prompts regarding a factoid question, can a large language model (LLM) reliably generate factually correct answers? Existing LLMs may generate distinct responses for different prompts. In this paper, we study the problem of quantifying knowledge contained in an LLM regarding a given set of facts. We propose KaRR, a statistical approach to assess factual knowledge for LLMs. The main idea is to estimate the ratio of LLM generating text corresponding to the answer entity given diverse prompts of the subject and the querying relation, versus it generating by random chances. Our assessment suite contains a comprehensive set of 994,123 entities and 600 relations, with 1,395,905 text aliases. We use our method to evaluate 20 LLMs of various sizes, including LLaMA, Alpaca, OPT, etc. Experiments show that our results have a strong correlation (0.43 Kendall's $\tau$) with the results of human assessment on LLMs. Our results reveal that the knowledge in LLMs with the same backbone architecture adheres to the scaling law, while tuning on instruction-following data sometimes compromises the model's capability to generate factually correct text reliably.

## 1  Introduction

Large language models (LLMs) have achieved impressive performance on many tasks including text generation, question answering, and dialog generation [Brown et al., 2020, Thoppilan et al., 2022, Bubeck et al., 2023]. Previous studies have shown that pretrained language models store factual knowledge in parameters [Roberts et al., 2020, Dai et al., 2022a,b, Singhal et al., 2023], demonstrating considerable potential as assistants for responding to user questions based on their stored knowledge.

Despite the remarkable success of LLMs, critical concerns arise — LLMs often generate unreliable answers given varying prompts [Ji et al., 2023, Maharana et al., 2023, Chang and Bergen, 2023, Chen et al., 2023]. As shown in Fig. 1, reliability refers to the ability to consistently generate knowledge-correct text, posing a higher standard than accuracy. For example, although Alpaca-7B [Taori et al., 2023] generates accurately (*playwright*) given a prompt of "*William Shakespeare's job*", it generates incorrect text (*boatman*) given a prompt of "*the job of Swan of Avon is*"[1]. ChatGPT [OpenAI, 2022] correctly generates *playwright and teacher*. However, it answers "No" for the prompt "*Is William Shakespeare a teacher?*" In this paper, we assess the knowledge contained in an LLM and investigate whether an LLM is able to consistently generate factually correct answers given varying prompts. Knowledge assessment is a critical problem for LLMs. The assessment results directly affect the people's trust in the LLM generated content. Once we identify inconsistency of LLM generation, we could potentially correct such knowledge in LLMs [De Cao et al., 2021, Zhao et al., 2021, Dong et al., 2022].

---

[1]Swan of Avon is a nickname of Shakespeare.

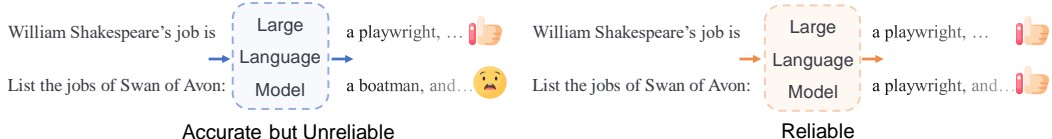

Figure 1: Accuracy pertains to generating the correct knowledge in response to a specific prompt, evaluated as a binary value (correct or incorrect). However, accuracy alone does not ensure reliability, as responses may vary with different prompt expressions. Reliability requires consistently generating correct knowledge for semantically similar prompts and is measured as a continuous value. Reliability sets a higher standard than accuracy by maintaining consistent accuracy across related prompts.

Important steps have been taken in the automatic knowledge evaluation for pre-trained language models [Petroni et al., 2019, Poerner et al., 2020, Jiang et al., 2020]. However, two issues still remain: 1) Prior methods focus on assessing the accuracy but not the reliability. Consequently, their probing results are prone to inflation or bias since models exhibit spurious correlations towards specific prompts [Poerner et al., 2020]. 2) Most prior methods target masked language models (MLMs) and don't provide a universal solution for assessing the knowledge in generative LLMs. MLMs are evaluated by probing whether they predict the masked object in response to a given prompt in cloze form, while the freely generated results of generative LLMs might be irrelevant to factual knowledge. Therefore, there is a need for a method that assesses both the accuracy and consistency, i.e. reliability, of knowledge in LLMs.

In this paper, we introduce KaRR, a statistical approach to assess whether a large language model contains reliable factual knowledge. We use triplets to represent the probing facts (e.g. `<William_Shakespeare, occupation, Playwright>`). Our main idea is to estimate the ratio of the likelihood of LLM generating correct surface text for ground-truth object entities (`Playwright`) given varying prompts with subject entity (`William_Shakespeare`) and relation (`occupation` and the likelihood of generating entity text by pure chances. We distinguish the entities versus their surface text since each entity and relation may contain several text aliases. We introduce three latent variables to represent subject, object, and relation, and a Bayesian network to represent the causal dependency among them and generated text. Our proposed KaRR quantifies the ratio of generating text with and without a specified relation/subject. This metric effectively captures the combined impact of both the subject and the relation in determining the model generation probability for the plausible text of the object (e.g., "playwright", "dramatist").

We develop a large-scale assessment suite with 994,123 entities, 600 relations, and their text aliases. Our suite is orders of magnitude larger than prior studies which only cover very few relations. We evaluate a comprehensive set of 20 LLMs using our method, including LLaMA [Touvron et al., 2023a], OPT [Zhang et al., 2022], and others. Moreover, we also invite human experts to assess knowledge in these LLMs. Results reveal a strong correlation (0.43 Kendall's $\tau$) between KaRR and human evaluation. Our experimental results show that knowledge of LLMs with the same backbone architecture conforms to the scaling law.

In summary, our contributions are: (1) We propose KaRR, a statistical method to assess the reliable knowledge contained in LLMs. We present a probabilistic graphical model to tackle varying text aliases regarding a fact. (2) We develop a large-scale assessment suite with 994,123 entities and 600 relations, which will be released to the community for further study. (3) We conduct a comprehensive evaluation of 20 LLMs. The experiments show that KaRR exhibits a strong correlation (0.43 Kendall's $\tau$) with human assessment, and achieves a low variance to varying prompts, with a standard deviation of 0.82. Furthermore, our evaluation results are seldom affected by spurious correlations. Our code and data are available at https://github.com/dqxiu/KAssess.

## 2 Related Work

**Knowledge Probing** Recent studies have demonstrated that language models possess factual knowledge, which can be utilized for complex reasoning, question answering, and performing various tasks[Roberts et al., 2020, Choudhary and Reddy, 2023]. To evaluate the knowledge in language models, previous works implement knowledge probing methods on MLMs. Petroni et al. [2019] introduces LAMA, studying whether MLMs correctly predict masked object entities in a cloze-style prompt. However, Elazar et al. [2021] shows that probing outcomes are inconsistent with various prompts, resulting in contradictory or unreliable results. Shin et al. [2020] show that

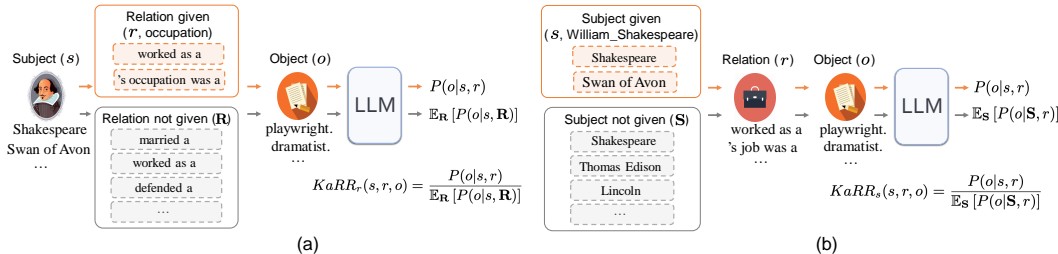

Figure 2: Illustration of KaRR using the fact `<William_Shakespeare,occupation,playwright>` as an example. $\mathbf{S}$ and $\mathbf{R}$ are latent variables representing the subject and relation, respectively; $s, r$ and $o$ refer to specific values of a subject, relation and object, respectively. (a) illustrates $\text{KaRR}_r$, which measures the impact of specifying $r$ or not on LLM generating $o$ given $s$. (b) illustrates $\text{KaRR}_s$, measuring the impact of specifying $s$ or not on LLM generating $o$ given $r$.

well-engineered automatic prompts result in much higher results than handcrafted LAMA prompts. Therefore, ParaRel [Elazar et al., 2021] and LPAQA [Jiang et al., 2020] are proposed for MLMs. ParaRel measures the knowledge consistency of MLMs through the use of several paraphrased prompts. Subsequently, Hase et al. [2023] adopt their metric for paraphrase consistency as a measure of belief, and Raj et al. [2022] extend the metric to measure the general semantic consistency of OPT [Zhang et al., 2022]. LPAQA focuses on prompts selection and ensembling to improve the prediction accuracy, by automatically discovering better prompts to query. However, it has not been comprehensively assessed to what extent LLMs contain knowledge about a fact.

**Spurious Correlations in Probing**   Recent literature reveals that empirical probing results for masked entities may be affected by spurious correlations within the text [Poerner et al., 2020, Dong et al., 2022]. Spurious correlations occur when a model takes shortcuts to generate answers based on superficial information in surface forms, even without possessing relevant knowledge. For instance, a model with limited knowledge can correctly answer "French" when queried about the native language of actor "Jean Marais", simply because "Jean Marais" is a typically French name. This phenomenon inflates probing results. On the one hand, Zhao et al. [2021] find that probing results are influenced by contextual biases, and Dong et al. [2022] also note frequency biases in rank-based probing. On the other hand, the subject's class has been shown to introduce biases in predictions [Poerner et al., 2020]. To address these issues, Poerner et al. [2020] eliminate vulnerable relation classes, however, this approach restricts coverage and does not provide a comprehensive solution. In our statistical knowledge assessment, we account for both the subject and relation by decomposing their influence and using multiple aliases to minimize spurious correlations.

# 3   Statistical Knowledge Assessment

In this section, we present a statistical method for assessing knowledge in LLMs. We begin by introducing the graphical model for knowledge assessment, which serves as the foundation of our approach. Subsequently, we introduce the Knowledge Assessment Risk Ratio (KaRR), which we then convert into a computable form based on the graphical model.

## 3.1   Graphical Model for Knowledge Assessment

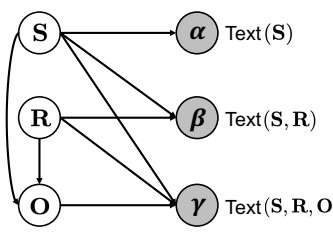

Figure 3: Graphical model for knowledge assessment.

Fig. 3 shows the graphical model that describes the knowledge symbols and their corresponding text forms. Here, we introduce latent variables $\mathbf{S}, \mathbf{R}$, and $\mathbf{O}$ to represent the symbolic subject, relation, and object, respectively. We use $s, r$, and $o$ to refer to specific values of these variables, and the triplet $(s, r, o)$ to denote a simple fact/knowledge, such as `<William_Shakespeare,occupation,playwright>`. We introduce $\boldsymbol{\alpha}, \boldsymbol{\beta}, \boldsymbol{\gamma}$ to denote the random variables for the textual forms (textual aliases[2]) of $(\mathbf{S}), (\mathbf{S}, \mathbf{R}), (\mathbf{S}, \mathbf{R}, \mathbf{O})$, respectively. Here, $\alpha, \beta$, and $\gamma$ refer to specific values of $\boldsymbol{\alpha}, \boldsymbol{\beta}$, and $\boldsymbol{\gamma}$, such as $\alpha$ being expressed as *"Shakespeare"*.

---

[2]"Aliases" are alternative names for entities or relations, defined in Wikidata (https://www.wikidata.org/wiki/Help:Aliases).

In general, fully observing the LLM probability on the latent variables of symbolic knowledge is infeasible. Although we can calculate the probability of the model generating *"Shakespeare"*, we cannot calculate the probability of it generating the symbol `<William_Shakespeare>`. This is because LLM pretraining focuses on textual input rather than symbolic representations, and each symbol may encompass a multitude of textual variations (paraphrased expressions) that signify the same entity or relation. Fortunately, we can establish a connection between symbols and text forms, enabling the calculation of the model's probability for specific textual forms. As demonstrated in the graphical model, a hollow circle indicates the variable is latent, and a shaded circle indicates the variable is observed. The parent node of $\alpha$ is $\mathbf{S}$, while $\mathbf{S}$ and $\mathbf{R}$ are parents to $\beta$. Additionally, $\gamma$ has three parent nodes, namely $\mathbf{S}$, $\mathbf{R}$, and $\mathbf{O}$. This structure enables the calculation of model probability on symbols through their corresponding text forms. Therefore, the goal of consistent LLM knowledge assessment is to ***estimate the model knowledge on symbols through the observable model probability across diverse corresponding textual forms***.

### 3.2 Knowledge Assessment Risk Ratio

For a given LLM $\mathcal{M}$, we define its factual knowledge correctness regarding a simple fact $(s, r, o)$ as the extent to which it generates knowledge-correct answers to various textual prompts. Specifically, we assess the joint impact of subject and relation symbols on the LLM's ability to generate the object symbol. This joint impact comprises two components: (1) the impact of specifying $r$ or not on $\mathcal{M}$ generating $o$ given $s$ and (2) the impact of specifying $s$ or not on $\mathcal{M}$ generating $o$ given $r$.

Risk Ratio (RR) is a statistical measure employed to compare the likelihood of a specific event occurring in an exposed group versus an unexposed group [Sistrom and Garvan, 2004]. To quantify the impact of specifying symbols (or not) on the LLM generation accuracy, we adapt RR into a new metric for knowledge assessment called KaRR. For component (1), we consider the exposed group as the case where a relation symbol $r$ is specified, and the unexposed group as cases where the relation is not specified. The outcome is the probability of model generating the object symbol $o$, representing any possible textual representation of $o$. We denote the RR of specifying relation $r$ as $\text{KaRR}_r$:

$$\text{KaRR}_r(s, r, o) = \frac{P(o|s, r)}{\mathbb{E}_{\mathbf{R}}\left[P(o|s, \mathbf{R})\right]} = \frac{P(o|s, r)}{P(o|s)} \tag{1}$$

As the example illustrated in Fig. 2 (a), for the fact `<William_Shakespeare,occupation, playwright>`, $\text{KaRR}_r$ compare the probability of model generating any alias of the object entity `playwright` when the relation is specified as `occupation` (e.g., *"Shakespeare works as a"*, *"William Shakespeare's occupation is a"*), with that when the relation is not specified, the expectation of model probability on generating an object alias under various possible relations such as *"Shakespeare married a"*. Similarly, the Risk Ratio (RR) for a specified subject $s$, denoted as $\text{KaRR}_s$, is defined as,

$$\text{KaRR}_s(s, r, o) = \frac{P(o|s, r)}{\mathbb{E}_{\mathbf{S}}\left[P(o|\mathbf{S}, r)\right]} = \frac{P(o|s, r)}{P(o|r)} \tag{2}$$

While the numerator for $\text{KaRR}_s$ is the same as that for $\text{KaRR}_r$, it compares the numerator with the model's probability when the subject is not specified. As shown in Fig. 2 (b), $\text{KaRR}_s$ calculates the expected model probability of generating an object alias across multiple possible subjects (e.g., *"Dante worked as a"* and *"Thomas Edison worked as a"*). To effectively represent the combined impact of $\text{KaRR}_s$ and $\text{KaRR}_r$, we calculate the joint influence as the geometric mean of them, and name it Knowledge Assessment Risk Ratio (KaRR). Formally, we define KaRR on $(s, r, o)$ as,

$$\text{KaRR}(s, r, o) = \sqrt{\text{KaRR}_r(s, r, o) \cdot \text{KaRR}_s(s, r, o)} \tag{3}$$

Intuitively, there are two components in KaRR: (1) the ratio of the chance of LLM generating $o$ given $s$ and $r$, and that of LLM generating $o$ only given $s$; (2) the ratio of the chance of M generating $o$ given $s$ and $r$, and that of LLM generating o only given $r$.

### 3.3 Computing KaRR using Graphical Model

The Knowledge Assessment Risk Ratio (KaRR), as indicated in Eq. (3), is formulated based on knowledge symbols, while directly computing the KaRR score using the definition (on symbols) is unfeasible. In light of this, our graphical model for knowledge assessment facilitates the implementation of KaRR by employing model probabilities on the text.

Based on Fig. 3, we can use the graphical model to evaluate the numerator of KaRR$_s$ and KaRR$_r$ in Eq. (1) and Eq. (2). To do so, we can use the following formula:

$$P(o|s,r) = \sum_{k=1}^{|\boldsymbol{\beta}|} P(o, \beta_k | s, r) = \sum_{k=1}^{|\boldsymbol{\beta}|} P(\beta_k | s, r) \cdot P(o | s, r, \beta_k) \qquad (4)$$

The probability is computed by summing over all possible values of $\boldsymbol{\beta}$, namely the set of aliases for $(s, r)$. For each specific value of $\boldsymbol{\beta}$, $\beta_k$, we apply the product rule of probability and decompose $P(o, \beta_k | s, r)$ into the probability of observing the relation $\beta_k$ given $s$ and $r$, denoted by $P(\beta_k | s, r)$, and the probability of observing the object $o$ given the subject $s$, relation $r$, and the specific relation type $\beta_k$, denoted by $P(o | s, r, \beta_k)$. We use $P_{\mathcal{M}}$ to denote the generation probability of model $\mathcal{M}$. According to the law of total probability for marginal distribution, $P(o | s, r, \beta_k)$ can be expanded as,

$$P(o | s, r, \beta_k) = \sum_{j=1}^{|\boldsymbol{\gamma}|} P(o, \gamma_j | s, r, \beta_k) = \sum_{j=1}^{|\boldsymbol{\gamma}|} P_{\mathcal{M}}(\gamma_j | s, r, \beta_k) P(o | \gamma_j) \qquad (5)$$

In the context of calculating the denominator of KaRR$_r$, it is important to note that each subject symbol may possess multiple aliases. For instance, the subject symbol $s$, `<William_Shakespeare>`, may be expressed with varying probabilities as $\alpha_1$ (*"Shakespeare"*), $\alpha_2$ (*"Swan of Avon"*), $\alpha_3$ (*"William Shakespeare"*), and so forth. To account for this variability, we expand the denominator as demonstrated in Eq. (6) by leveraging principles of marginal distribution and Bayes Rule.

$$P(o|s) = \sum_{i=1}^{|\boldsymbol{\alpha}|} P(o, \alpha_i | s) = \sum_{i=1}^{|\boldsymbol{\alpha}|} P(\alpha_i | s) P(o | s, \alpha_i) \qquad (6)$$

$P(o | s, \alpha_i)$ is obtained by summing the joint probabilities of $o$ and $\gamma_j$ given $s, \alpha_i$ under all possible $\gamma_j$. Mathematically, this can be expressed as:

$$P(o | s, \alpha_i) = \sum_{j=1}^{|\boldsymbol{\gamma}|} P(o, \gamma_j | s, \alpha_i) = \sum_{j=1}^{|\boldsymbol{\gamma}|} P_{\mathcal{M}}(\gamma_j | s, \alpha_i) P(o | s, \alpha_i, \gamma_j) \qquad (7)$$

Subsequently, we obtain $RR_r(s, r, o)$ as follows,

$$\text{KaRR}_r(s, r, o) = \frac{P(o|s,r)}{P(o|s)} = \frac{\sum_{k=1}^{|\boldsymbol{\beta}|} P(\beta_k | s, r) \sum_{j=1}^{|\boldsymbol{\gamma}|} P_{\mathcal{M}}(\gamma_j | s, r, \beta_k) P(o | \gamma_j)}{\sum_{i=1}^{|\boldsymbol{\alpha}|} P(\alpha_i | s) \sum_{j=1}^{|\boldsymbol{\gamma}|} P_{\mathcal{M}}(\gamma_j | s, \alpha_i) P(o | s, \alpha_i, \gamma_j)} \qquad (8)$$

For the denominator of KaRR$_s$, in the above example, $P(o|r)$ represents the expected probability of generating an alias for `playwright` when presented with a sentence that semantically conveys the `occupation` relation, over all the subject symbols. Consequently, we can expand $P(o|r)$ as $\sum_{u=1}^{|\boldsymbol{s_u}|} P(s_u | r) P(o | s_u, r)$. The computation of $P(o | s_u, r)$ is identical to that in Eq. (4). Combining these results, we can determine the value of KaRR$_s(s, r, o)$ as Eq. (9).

$$\text{KaRR}_s(s, r, o) = \frac{\sum_{k=1}^{|\boldsymbol{\beta}|} P(\beta_k | s, r) \sum_{j=1}^{|\boldsymbol{\gamma}|} P_{\mathcal{M}}(\gamma_j | s, r, \beta_k) P(o | \gamma_j)}{\sum_{u=1}^{|\boldsymbol{s_u}|} P(s_u | r) \cdot \sum_{k=1}^{|\boldsymbol{\beta}|} P(\beta_k | s_u, r) \sum_{j=1}^{|\boldsymbol{\gamma}|} P_{\mathcal{M}}(\gamma_j | s_u, r, \beta_k) P(o | \gamma_j)} \qquad (9)$$

The KaRR score is calculated as the geometric mean of the results obtained from Eq. (8) and (9), and its detailed implementation is shown in Appendix A.

## 4 Experiments

We implement KaRR to examine knowledge in 14 LLMs and analyze its correlation with human assessment on model knowledge. Compared to large language models, knowledge graphs store a vast amount of structured information explicitly, which is ensured to be correct through manual construction and verification. Therefore, we implement KaRR based on existing large knowledge graphs, including T-REx [Elsahar et al., 2018] and Wikidata [Vrandečić and Krötzsch, 2014].

### 4.1 Data and Settings

In the following paragraphs, we provide details on the data, models and settings used in our experiments, with additional information provided in Appendix B and C.

**Symbolic Facts** We utilize T-REx knowledge graph [Elsahar et al., 2018] as our primary source of symbolic knowledge. T-REx comprises 11 million triples that are aligned with 3.09 million Wikipedia abstracts; its quality is validated by extensive crowdsourcing evaluation. In our main experiments, we consider all 600 English relations available in T-REx and sample a maximum of 20 facts per relation, resulting in a total of 10,691 facts for knowledge assessment.

| Method | Subj. Alias | Obj. Alias | Rel. Alias | Rel. Cvg. |
|---|---|---|---|---|
| LAMA@1 | ✗ | ✗ | ✗ | 6.83% |
| LAMA@10 | ✗ | ✗ | ✗ | 6.83% |
| ParaRel | ✗ | ✗ | ✓ | 6.33% |
| KaRR | ✓ | ✓ | ✓ | 100% |

| Method | Recall | Kendall's $\tau$ | p-value |
|---|---|---|---|
| LAMA@1 | 83.25% | 0.17 | 0.10 |
| LAMA@10 | 65.81% | 0.08 | 0.23 |
| ParaRel | 69.15% | 0.22 | 0.02 |
| K-Prompts | 78.00 % | 0.32 | 0.03 |
| KaRR | **95.18%** | **0.43** | 0.03 |

(a) Basic information.    (b) Human evaluation results.

Table 1: Basic information and human evaluation results. The abbreviations Subj., Rel., Obj., and Rel. Cvg. represent Subject, Relation, Object, and the coverage of English relations in T-REx, respectively. The methods evaluated through human evaluation are assessed using the same set of 410 randomly sampled facts. K-Prompts is implemented using shared basic information with KaRR.

**Multiple Entity Aliases** For the text forms of subjects and objects involved in calculating $KaRR_s$ and $KaRR_r$, we search the entity aliases from Wikidata with Wikidata Integrator[3]. Ultimately, we obtain 1,349,474 aliases for 968,426 subjects and 368,511 aliases for 207,985 objects.

**Multiple Relation Templates** Previously, Elazar et al. [2021] manually wrote paraphrased relation templates for 41 relations, covering 6% of the relation classes in T-REx. To expand the coverage and diversity of templates for **R**, we incorporated placeholders "[X]" and "[Y]" into the relation aliases in Wikidata. Subsequently, we utilized a fine-tuned Flan-T5 model[4] for grammar correction to generate corresponding template candidates. We then filtered the templates with missing subjects or objects, and the remaining template candidates were all validated and corrected manually by two human annotators. Finally, we obtained 3,488 templates for 600 relations, with an average of 5.82 paraphrased templates per relation, achieving full coverage of the English relation classes in T-REx.

**Overall KaRR Score** The KaRR score assigned to each fact is a continuous value. If the KaRR score exceeds a predefined threshold, the fact is considered consistently known by the LLM. When applying our method to a set of facts for the general knowledge reliability of LLMs, the overall KaRR score is calculated as the proportion of facts with a KaRR score exceeding the threshold.

As the statistical information shown in Tab. 1 (a), our method exhibits a broader coverage of relations and entity aliases compared to previous studies. To implement KaRR, we initially sample the knowledge triplet from T-REx, which comprises the symbolic subject $s$, relation $r$, and object $o$. By leveraging entity aliases and relation templates that we have devised, we can generate multiple expressions for each element in the KaRR metric. For instance, $\alpha$ is implemented as the set of all searched subject aliases from our aliases datastore, and $\beta$ represents the combination of all possible relation templates and subject aliases (by replacing the placeholder with the subject aliases).

### 4.2 Models

We explore the following models for knowledge assessment: (1) traditional size pretrained models ($< 5B$), including GPT [Radford et al., 2018], XLNet [Yang et al., 2019], GPT2-XL [Radford et al., 2019], GPT-NEO [Black et al., 2021], T5-large and T5-3B [Raffel et al., 2020] and Phi-1.5 [Li et al., 2023] ; (2) medium size LLMs ($\geq 5B$, $< 50B$), including GLM [Du et al., 2022], Falcon [Penedo et al., 2023], Dolly[5], Moss[6], LLaMA [Touvron et al., 2023a] and its variants Alpaca [Taori et al., 2023], Vicuna [Chiang et al., 2023] and WizardLM [Xu et al., 2023]; (3) large LLMs ($\geq 50B$), including the 65B LLaMA [Touvron et al., 2023a], the 65B LLaMA2 [Touvron et al., 2023b] and the 175B OPT [Zhang et al., 2022].

### 4.3 Results of Knowledge Assessment

We present the results of our statistical knowledge assessment on various LLMs in Tab. 4 and more detailed results in Appendix D. Among the small and the medium-sized LLMs, most models' KaRR scores are between 3 to 13, indicating that they often struggle to generate factually accurate sentences consistently. However, it is surprising to note that the 2.65B GPT-NEO outperforms the T5-3B with a 3.92 KaRR score difference. We hypothesize that this difference in performance may be attributed to the pretraining corpora. Specifically, GPT-NEO (trained on the Pile dataset [Gao et al., 2021])

---

[3] https://github.com/SuLab/WikidataIntegrator
[4] https://huggingface.co/pszemraj/flan-t5-large-grammar-synthesis
[5] https://github.com/databrickslabs/dolly
[6] https://github.com/OpenLMLab/MOSS

| Model | Size | KaRR Score | Model | Size | KaRR Score |
|-------|------|-----------|-------|------|-----------|
| GPT | 0.12B | 9.57 | GLM | 10B | 5.59 |
| XLNet | 0.12B | 5.86 | Dolly | 12B | 15.60 |
| T5-large | 0.74B | 3.22 | LLaMA | 13B | 13.86 |
| Phi-1.5 | 1.3B | 10.58 | Alpaca | 13B | 8.24 |
| GPT2-XL | 1.56B | 12.27 | Vicuna | 13B | 19.50 |
| GPT-NEO | 2.65B | 13.44 | WizardLM | 13B | 16.90 |
| T5-3B | 3B | 9.52 | Moss | 16B | 11.20 |
| Falcon | 7B | 7.97 | LLaMA | 65B | 14.56 |
| BLOOM | 7B | 7.72 | LLaMA2 | 65B | 19.71 |
| LLaMA | 7B | 12.37 | OPT | 175B | 23.06 |

Table 2: Statistical knowledge assessment results on 20 LLMs. KaRR scores indicate that fine-tuning on instruction-following data (Alpaca) jeopardizes the reliability of knowledge-correct generation of the original LLaMA, whereas utilizing data from a more knowledgeable model enhances model knowledge (Vicuna). Additionally, the same backbone architecture adheres to the scaling law.

primarily utilizes high-quality data, while T5 (pretrained on a combination of corpora ) uses a more varied set of data sources. For larger LLMs, a comparison between the original LLaMA and the Alpaca model reveals that instruction-tuning might influence the model's ability to generate consistent and correct knowledge. Compared to the KaRR score of the original LLaMA, the KaRR score of Vicuna indicates that fine-tuning LLM on data collected from a more knowledgeable model could augment its knowledge. This finding is not commonly observed in previous studies but is consistent with our case study presented in Appendix E.

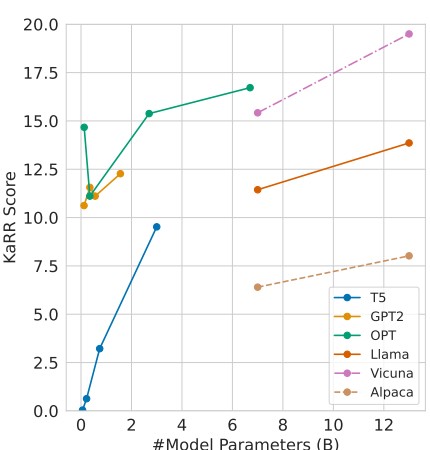

Figure 4: Knowledge in models with the same backbone architecture follows the scaling law.

Fig. 4 demonstrates that our statistical evaluation approach supports the law of model scaling, where larger models tend to possess more factual knowledge. This observation is consistent with our intuition about the relationship between model size and knowledge representation. However, the benefits of scaling vary across models. For instance, T5 shows a significant improvement in KaRR score when scaled from T5-small to T5-3B, while GPT2 and OPT also benefit from scaling, albeit to a lesser extent. Notably, we observe that when the model size is smaller than 1B, GPT2 and OPT significantly outperform T5 in generating correct knowledge. This finding highlights the importance of selecting an appropriate model size for a given task and dataset, as larger models may not always provide the best performance on generation factuality. Overall, the results of our statistical evaluation method are consistent with our intuition, and highlight the importance of large-scale pretraining on knowledge-rich corpora for building high-performing language models.

## 4.4 Human Evaluation for Knowledge Assessment

To further validate our approach, we conduct the human evaluation for model knowledge assessment. Utilizing the mean score from human evaluation as the benchmark for knowledge assessment on individual facts, we examine the validity of our knowledge assessment outcomes from two perspectives: 1) the $\tau$ [Kendall, 1938] correlation between the KaRR score on each fact and the human-annotated score on each fact and 2) the Recall for identifying knowledge that is not consistently known (facts with human average scores below 0.5) when employing the GPT2-XL model.

**Human Evaluation**  Our human evaluation for the reliable knowledge generation ability of GPT2-XL contains two procedures: 1) Annotating: We have three annotation volunteers with NLP backgrounds. Each annotator is asked to write three prompts to probe the model knowledge from multiple aspects for each fact. They are instructed to refine the prompts based on model generations until the generations are of the same type as the target answer (but not necessarily the target entity). To ensure a fair comparison with LAMA [Petroni et al., 2019], which is only applicable to 41 high-frequency

| Method | Var (↓) | Std (↓) |
|--------|---------|---------|
| LAMA@1 | 1.90 | 1.37 |
| LAMA@10 | 5.14 | 2.27 |
| ParaRel | 0.77 | 0.94 |
| K-Prompts | 2.34 | 5.47 |
| KaRR | **0.67** | **0.82** |

| Method | SP (↓) | ΔP (↓) |
|--------|--------|--------|
| LAMA@1 | 3.81 | 0.00 |
| LAMA@10 | 64.29 | 47.31 |
| ParaRel | 2.66 | -0.51 |
| K-Prompts | **0.00** | -7.54 |
| KaRR | 1.94 | **-14.94** |

(a) Evaluation variance towards varied prompts.  (b) Spurious correlation of knowledge assessment.

Table 3: Evaluation variance and spurious correlation analysis. Compared to LAMA and ParaRel, our statistical knowledge assessment is more robust and less influenced by the spurious correlation.

relations, we restricted the relation classes to those included in LAMA. In total, we obtained 4,140 valid manually annotated prompts for 410 facts in T-REx. 2) Rating: We then asked three other annotators to rate the knowledge (0 or 1) in GPT2-XL according to the model generations. We used the average score of their ratings on all the prompts as the gold standard for knowledge assessment on each fact. Detailed annotation procedures and scoring criteria are shown in Appendix F.

**Baselines**    As a comparison, we also calculated the correlation between three baseline methods and human evaluation results. The baseline methods, LAMA [Petroni et al., 2019] and ParaRel [Elazar et al., 2021], were designed for MLMs, we modified them for LLMs by deleting words following the objects in the templates. We implement the *Consistent-Acc* score [Elazar et al., 2021] for ParaRel, which combines knowledge and consistency during evaluation. LAMA@1 and LAMA@10 refer to the results obtained by checking whether the top-1 or top-10 generations contain the object, respectively. Furthermore, we propose a K-Prompts baseline that computes the average probability of k randomly sampled prompts. It can be considered an adapted version of LPAQA [Shin et al., 2020] for LLMs, utilizing a collection of ensembled high-quality prompts identical to those used in KaRR. Here, we adopt K-Prompts as a simple baseline to evaluate the knowledge assessment in the absence of latent variables. All baseline methods are executed on the same set of facts as KaRR. LAMA and ParaRel are implemented as per the original papers [Petroni et al., 2019, Elazar et al., 2021], and K-Prompts is implemented using shared aliases and templates with KaRR. We set 22 as the threshold of KaRR, which is chosen by aligning the proportion of GPT2-XL known facts distinguished by humans on a sampled set of 200 facts. Similarly, the threshold for K-Prompts has been set to 0.13, as a result of aligning with the proportion recognized by humans.

**Results**    Tab. 1 (b) demonstrate that KaRR performs exceptionally well in knowledge detection, achieving a Recall of 95.18%. This highlights the potential of our statistical knowledge assessment to identify unreliable knowledge in LLMs, which is essential for downstream knowledge updating [De Cao et al., 2021]. Furthermore, the evaluation results of KaRR show a strong correlation (Kendall's 0.43) with human evaluation results. However, averaging the results of paraphrased prompts (ParaRel and K-Prompts) is not effective due to varying prompt frequencies and diverse impacts of different prompts on the model's probability of generating an "object". Since human annotation for model knowledge assessment is demanding and time-consuming, our statistical evaluation method provides a fast and accurate solution for the automatic knowledge assessment of LLMs.

## 5    Variance toward Prompts

Since the aim of the evaluation is to reflect the correctness of an LLM's knowledge towards various prompts, the evaluation result should be stable and not easily affected by the text forms used. In this section, we investigate the evaluation variance of our method. In contrast to ParaRel [Elazar et al., 2021], which studies the consistency of models under paraphrased inputs, we investigate the variance of the knowledge assessment method. Specifically, we evaluate the evaluation variance, which measures how much the same evaluation results vary when different prompts are used.

Following the same procedures as described in Sec. 4.4, we sampled 4,100 facts and employed the paraphrased relation templates from ParaRel [Elazar et al., 2021] to replace the relation templates in the baseline methods and KaRR, resulting in three distinct relation templates for each fact. We evaluated the effectiveness of the different evaluation approaches on each of the three paraphrased relation templates separately, the model is GPT2-XL. Additionally, we measured the evaluation variance by calculating the variance and standard deviation of the evaluation scores.

Tab. 3 (a) illustrates that the results of LAMA and K-Prompts exhibit significant variations when the probing prompt is changed to a paraphrased one, despite the overall prompt similarity in ParaRel

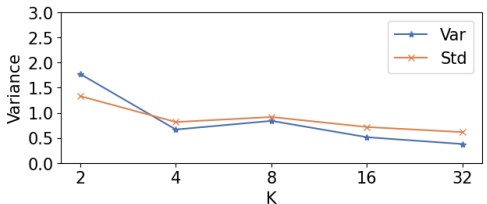 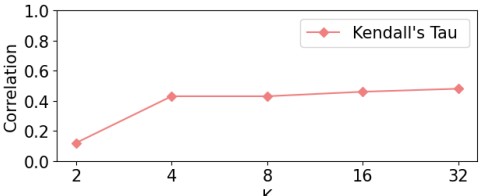

Figure 5: Ablation study on the sampling parameter K. With the increase in K, the variability in knowledge assessment decreases, while its correlation with human evaluation improves.

being $0.96$. In contrast, our statistical knowledge assessment approach demonstrates robustness to prompt variation, with a $0.67$ score variance and a $0.82$ standard deviation.

# 6 Spurious Correlation in Knowledge Assessment

Both LAMA and KaRR use text as a proxy to estimate unobservable knowledge. However, spurious correlations can lead to inflated probing results where LLMs "guess" the probing answer based on these correlations. For example, when probed with the relation template "[X]'s birthplace is [Y]", LLMs assign "America" the highest probability, regardless of the subject entity [Zhao et al., 2021]. To address this issue, we investigate spurious correlations in knowledge assessment, particularly focusing on the correlation between relation templates and high-frequency objects.

We synthesize false facts by replacing objects with high-frequency but factually incorrect ones. They are non-existent knowledge in LLMs as they are not part of the pretraining corpus. Ideally, the scores on synthetic facts should be 0 if the assessment is not influenced by spurious correlations. We obtain high-frequency objects by identifying the top-5 GPT2-XL predictions on 3 ParaRel relation templates without subjects, yielding 14 relations with valid high-frequency objects out of 41 relations. For each relation, we randomly sample 100 facts and replace the original object with the high-frequency object (if the original object is not the high-frequency object). We employ two metrics: Spurious Positive (SP) and Positive Gap ($\Delta$P). SP quantifies the proportion of false synthetic facts that are incorrectly recognized as known by the LLMs, and $\Delta$P is calculated by SP minus model's true positive rate on the actual facts. Ideally, SP and ($\Delta$P) should be close to 0 and ($\Delta$P) should be negative; otherwise, it indicates a significant bias in the evaluation result due to spurious correlations.

Tab. 3 (b) presents results on GPT2-XL. LAMA scores are highly correlated to the prompt, leading to the high $\Delta$P and SP. For example, $3.81\%$ and $64.29\%$ false facts are estimated as learned facts by LAMA@1 and LAMA@10, respectively. In contrast, K-Prompts and KaRR are less influenced by spurious correlations, with an SP score lower than $2\%$ and a negative $\Delta$P. By assessing the model knowledge from various perspectives, the ultimate scores of KaRR indicate the comprehensive knowledge mastery of LLMs. Therefore, our results are less influenced by co-occurrence shortcuts.

# 7 Discussion

In this section, we discuss the causal explanation of KaRR and explore the effects of the sampling parameter, as well as the influence of model size.

**Causal Explanation for KaRR** From a causal perspective, our problem can be defined as: *"given a subject $s$, the causal effect of given the relation $r$ or not for the LLM generating the object $o$"*. Assuming there are no confounders, given a fixed $s$, the treatment (T) is whether a specific $r$ is provided or not, the outcome (Y) is the probability of the model generating $o$, then the Average Treatment Effect (ATE) [Holland, 1986] is defined as:

$$\mathbb{E}[Y(1)] - \mathbb{E}[Y(0)] = \mathbb{E}[Y \mid T = 1] - \mathbb{E}[Y \mid T = 0] = P(o|s,r) - \mathbb{E}_{\mathbf{R}}\left[P(o|s,\mathbf{R})\right] \quad (10)$$

Note that in ATE, the minuend and subtrahend correspond to the denominators and numerator of $\text{KaRR}_r$ in Eq. (1), respectively. Assuming no confounders are present, the only distinction lies in the mathematical operation utilized (subtraction or division). Both subtraction and division can be used to compare the difference between the exposed group and the unexposed group, with the only distinction being the difference in the threshold for the resulting value. As a result, ATE helps clarify and validate KaRR from a causal perspective.

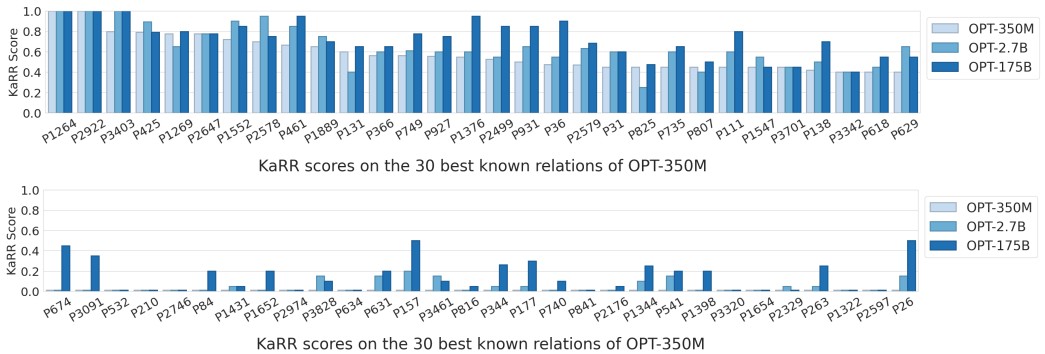

Figure 6: KaRR scores on different relations when scaling up OPT from 350M to 2.7B and 175B.

**Influence of Sampling Parameter** When calculating KaRR, the sampling number K makes a trade-off between the evaluation accuracy and efficiency. We perform an ablation study on the results of KaRR implemented with different K. Following the same settings of Sec. 5 and Sec. 4.4, we analyze the evaluation variance and evaluation accuracy of the overall KaRR scores of GPT2-XL. Fig. 5 demonstrates that increasing K decreases variance in knowledge assessment, but the impact is minor since evaluation variance is already low. The sampling parameter has a more significant effect on evaluation accuracy, with higher accuracy correlating with higher K values. When K is less than 4, the evaluation results are unreliable due to weak correlation with human evaluation results.

**Impact of Scaling on LLM Knowledge** To further study the impact of scaling up on LLM knowledge, we compare the KaRR scores of OPT-350M, OPT-2.7B, and OPT-175B on 30 best-known and 30 worst-known relations of OPT-350M. Fig. 6 shows that larger models exhibit better and more consistent knowledge-correct generation ability on the relations where small models also show some knowledge-correct generation ability. Moreover, larger models surpass small models in terms of knowledge on a wider range of relations. Therefore, the scaling law of LLM knowledge is attributed to the broader and more extensive knowledge of larger LLMs.

## 8 Limitations and Future Work

While KaRR demonstrates promising effectiveness for LLM knowledge assessment, there are a few aspects that require further investigation. First, KaRR scores are calculated for each fact using LLM inference on various prompts, which can be time-consuming and may limit the scalability of the evaluation. Second, our assessment is currently limited to scenarios where the logits of LLMs are available. This excludes many real-world applications where the models' internal representations may not be accessible or may be difficult to obtain. Third, the generality of KaRR can be further expanded. The current implementation focuses on atomic knowledge, characterized by triplets in the format of <subject, relation, object>. Nevertheless, real-world scenarios encompass a significant portion of complex knowledge, necessitating multi-hop reasoning or the comprehension of intricate relationships among multiple entities [Choudhary and Reddy, 2023]. Evaluating such knowledge in LLMs continues to be a challenging and unresolved issue. Overall, these areas for improvement emphasize the opportunities for continued research and development of knowledge assessment approaches for LLMs, taking into account the complexities of real-world applications and the diverse types of knowledge to be assessed.

## 9 Conclusion

We investigate the consistent knowledge-correct generation ability of LLMs and propose a statistical knowledge assessment approach, called KaRR. By implementing our approach on 20 LLMs, we carry out an extensive assessment of the reliability of factual knowledge generation in these models. Our experiments reveal that our method yields a high correlation with human evaluation, and addresses the issues of variance and spurious correlation during the knowledge assessment of generative LLMs. Additionally, our assessment suite includes millions of entity and relation aliases, enabling large-scale knowledge evaluation. Overall, our method evaluates the reliable knowledge generation ability of LLMs, providing a robust foundation for future developments in knowledge refinement, augmentation, and beyond.

## Acknowledgement

We thank all the anonymous reviewers for their constructive comments, Yuxuan Fan, Lei Li and Ce Zheng for their valuable suggestions on our data collection. Zhifang Sui is the corresponding author. This paper is supported by the National Key Research and Development Program of China 2020AAA0106700 and NSFC project U19A2065.

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

# Appendix

## A  KaRR Implementation

To implement KaRR, we define a binary function $\delta_{text}(symbols)$ that takes the symbols as input and returns a value of either 0 or 1 depending on whether any alias of all the symbols appears in the text. By applying this function, we can transform $P(\beta_k|s,r)$ and $P(\alpha_i|s)$ into a format that is compatible with the model probability.

$$P(\beta_k|s,r) = \frac{P(\beta_k,s,r)}{P(s,r)} = \frac{P_{\mathcal{M}}(\beta_k) \cdot \delta_{\beta_k}(s,r)}{P(s,r)} \tag{11}$$

$$P(\alpha_i|s) = \frac{P(\alpha_i,s)}{P(s)} = \frac{P_{\mathcal{M}}(\alpha_i) \cdot \delta_{\alpha_i}(s)}{P(s)} \tag{12}$$

If the aliases $\beta_k$ contain any alias of both $s$ and $r$, then $P_{\mathcal{M}}(\gamma_j|s,r,\beta_k)$ is equal to $P_{\mathcal{M}}(\gamma_j|\beta_k)$. Otherwise, $P_{\mathcal{M}}(\gamma_j|s,r,\beta_k)$ is zero. Similarly, if the aliases $\alpha_i$ contain any alias of $s$, then $P_{\mathcal{M}}(\gamma_j|s,\alpha_i)$ is equal to $P_{\mathcal{M}}(\gamma_j|\alpha_i)$; otherwise, $P_{\mathcal{M}}(\gamma_j|s,\alpha_i)$ is zero.

Furthermore, the probabilities $P(o|\gamma_j)$ and $P(o|s,\alpha_i,\gamma_j)$ are both equal to $\delta_{\gamma_j}(o)$, where $\delta_{\gamma_j}$ is the binary function that checks whether any alias of the object $o$ appears in the text $\gamma_j$. By substituting the transformed formulas into the equation Eq. (9), we obtain an implementation of KaRR:

$$\text{KaRR}_r(s,r,o) = \frac{P(s) \sum_{k=1}^{|\boldsymbol{\beta}|} P_{\mathcal{M}}(\beta_k) \cdot \delta_{\beta_k}(s,r) \sum_{j=1}^{|\boldsymbol{\gamma}|} P_{\mathcal{M}}(\gamma_j|\beta_k)\delta_{\gamma_j}(o)}{P(s,r) \sum_{i=1}^{|\boldsymbol{\alpha}|} P_{\mathcal{M}}(\alpha_i) \cdot \delta_{\alpha_i}(s) \sum_{j=1}^{|\boldsymbol{\gamma}|} P_{\mathcal{M}}(\gamma_j|\alpha_i)\delta_{\gamma_j}(o)} \tag{13}$$

Based on the graphical model depicted above, $\mathbf{S}$ and $\mathbf{R}$ are conditionally independent when $\mathbf{O}$ is unknown. Consequently, we can simplify the expression $\frac{P(s)}{P(s,r)}$ to $\frac{1}{P(r)}$. Therefore,

$$\text{KaRR}_r(s,r,o) = \frac{1}{P(r)} \cdot \frac{\sum_{k=1}^{|\boldsymbol{\beta}|} P_{\mathcal{M}}(\beta_k) \cdot \delta_{\beta_k}(s,r) \sum_{j=1}^{|\boldsymbol{\gamma}|} P_{\mathcal{M}}(\gamma_j|\beta_k)\delta_{\gamma_j}(o)}{\sum_{i=1}^{|\boldsymbol{\alpha}|} P_{\mathcal{M}}(\alpha_i) \cdot \delta_{\alpha_i}(s) \sum_{j=1}^{|\boldsymbol{\gamma}|} P_{\mathcal{M}}(\gamma_j|\alpha_i)\delta_{\gamma_j}(o)} \tag{14}$$

The calculation of subject replacement is analogous. When $\mathbf{O}$ is unknown, $\frac{1}{P(s_u|r) \cdot P(s,r) \cdot \frac{1}{P(s_u,r)}}$ equals to $\frac{1}{P(s)}$, then we have,

$$\begin{aligned}
\text{KaRR}_s(s,r,o) &= \frac{\sum_{k=1}^{|\boldsymbol{\beta}|} P_{\mathcal{M}}(\beta_k) \cdot \delta_{\beta_k}(s,r) \sum_{j=1}^{|\boldsymbol{\gamma}|} P_{\mathcal{M}}(\gamma_j|\beta_k)\delta_{\gamma_j}(o)}{\sum_{u=1}^{|\boldsymbol{s_u}|} P(s_u|r) \cdot P(s,r) \sum_{k=1}^{|\boldsymbol{\beta}|} P_{\mathcal{M}}(\beta_k) \frac{1}{P(s_u,r)} \cdot \delta_{\beta_k}(s_u,r) \sum_{j=1}^{|\boldsymbol{\gamma}|} P_{\mathcal{M}}(\gamma_j|\beta_k)\delta_{\gamma_j}(o)} \\
&= \frac{1}{P(s)} \cdot \frac{\sum_{k=1}^{|\boldsymbol{\beta}|} P_{\mathcal{M}}(\beta_k) \cdot \delta_{\beta_k}(s,r) \sum_{j=1}^{|\boldsymbol{\gamma}|} P_{\mathcal{M}}(\gamma_j|\beta_k)\delta_{\gamma_j}(o)}{\sum_{u=1}^{|\boldsymbol{s_u}|} \sum_{k=1}^{|\boldsymbol{\beta}|} P_{\mathcal{M}}(\beta_k) \cdot \delta_{\beta_k}(s_u,r) \sum_{j=1}^{|\boldsymbol{\gamma}|} P_{\mathcal{M}}(\gamma_j|\beta_k)\delta_{\gamma_j}(o)}
\end{aligned} \tag{15}$$

In practical scenarios, computing all possible values of $\boldsymbol{\alpha}$ or $\boldsymbol{\beta}$ for the KaRR metric can be prohibitively expensive. To address this challenge, we can make the assumption that the symbols $\mathbf{S}$ and $\mathbf{R}$ follow a unified distribution, and then use Markov Chain Monte Carlo (MCMC) sampling to estimate the denominators by sampling K subjects or objects. This results in $\frac{1}{P(s_u)}$ or $\frac{1}{P(r_u)}$ being added to the denominators, offset by the $\frac{1}{P(r)}$ and $\frac{1}{P(s)}$ terms in Eq. (14) and (15). By employing this approach, we are able to improve the accuracy of the KaRR metric within a limited sampling budget and derive a new score based on the sampled subset. Our default sampling parameter, K, is set to 4.

The KaRR score is calculated as the geometric mean of the results obtained from Eq. (14) and (15).

## B  Experimental Details

In this section, we provide further information on the experimental details.

### B.1 Settings for Different LLMs

In this paper, we load the models (except the OPT-175B) from the Huggingface Models[7] and implement the OPT-175B according to Metaseq[8]. The specific version of Vicuna employed in this paper is lmsys/vicuna-7b-delta-v1.1.

### B.2 OOV Objects

We collect a vast range of aliases for objects, certain aliases may contain out-of-vocabulary (OOV) words. Consequently, the model generation probability for these aliases cannot be computed. To address this issue, we follow the approach suggested in LAMA [Petroni et al., 2019] and exclude OOV object aliases when implementing KaRR on each LLM.

### B.3 Scoring Details

According to the KaRR implementation above, we can obtain the KaRR score on each fact. If the KaRR is higher than a threshold, the fact is regarded as a consistently known one by the LLM; otherwise not. When implementing the statistical framework on a large set of facts for the general knowledge correctness of LLMs, we get the proportion of known facts over the 10,691 symbolic facts as the (KaRR higher than the threshold) as the overall KaRR score. We set 22 as the threshold for KaRR, and the threshold is chosen by aligning the proportion of GPT2-XL known facts distinguished by humans on a sampled set of facts. Also as a result of alignment with human-recognized proportion, the threshold for the K-Prompts baseline implemented in this paper is set to 0.13.

## C  Data Distribution

In this section, we provide detailed information on alias distribution and relation distribution.

### C.1  Alias Distribution

| Type | Top-k | Avg. Aliases | Proportion of Related Facts |
|---|---|---|---|
| subject | 50 | 7.62 | 12.10% |
| | 100 | 6.73 | 16.13% |
| | 500 | 4.68 | 28.34% |
| | 1000 | 3.70 | 33.70% |
| object | 50 | 5.06 | 39.84% |
| | 100 | 4.79 | 47.35% |
| | 500 | 3.80 | 65.44% |
| | 1000 | 3.48 | 71.93% |

Table 4: Average number of aliases and proportion of related facts for top-k frequent entities.

In Tab. 4, we present the average number of aliases and the proportion of related facts for the top-k frequent entities. The most frequent entities typically possess more than five aliases, related to a large proportion of facts. This long-tailed pattern is consistent with real-world alias distribution, where a substantial portion of entities have only one alias.

### C.2  Relation Distribution

As shown in Tab. 5, the distribution of relations in our dataset exhibits a long-tail phenomenon, which is consistent with the distribution of relations in real-world knowledge and the distribution of relations within knowledge graphs [Elsahar et al., 2018].

---

[7] https://huggingface.co/models
[8] https://github.com/facebookresearch/metaseq/tree/main

| Top-k | Avg. Aliases | # Related Facts | Proportion of Related Facts |
|---|---|---|---|
| 50 | 6.96 | 12,638,356 | 90.05% |
| 100 | 5.97 | 13,538,967 | 96.47% |
| 150 | 5.54 | 13,802,467 | 98.35% |
| 200 | 5.22 | 13,921,742 | 99.20% |
| 250 | 4.84 | 13,977,899 | 99.60% |

Table 5: Distributions of top-k relations.

| Model | Model Size | KaRR Score | KaRR$_s$ Score | KaRR$_r$ Score | Best-known Relations | Worst-known Relations |
|---|---|---|---|---|---|---|
| GPT | 0.12B | 9.57 | 4.30 | 31.9 | P2822, P1537, P3095 | P1427, P1951, P3091 |
| XLNet | 0.12B | 5.86 | 0.25 | 37.78 | P1537, P1264, P2922 | P664, P31, P1951 |
| T5-large | 0.74B | 3.22 | 4.61 | 6.82 | P872, P560, P2860 | P664, P1951, P3091 |
| Phi-1.5 | 1.3B | 10.58 | 6.30 | 36.67 | P3403, P538, P2500 | P1542, P184, P926 |
| GPT2-XL | 1.56B | 12.27 | 12.12 | 30.38 | P2922, P2499, P397 | P3091, P2159, P210 |
| GPT-NEO | 2.65B | 13.44 | 12.97 | 29.15 | P1537, P2499, P3403 | P210, P2746, P681 |
| T5-3B | 3B | 9.52 | 3.59 | 23.04 | P3095, P3033, P1537 | P1427, P674, P1951 |
| Falcon | 7B | 7.97 | 10.44 | 23.73 | P3403, P1376, P1165 | P805, P1542, P184 |
| BLOOM | 7B | 7.72 | 6.58 | 33.42 | P3403, P2922, P2289 | P263, P1542, P184 |
| LLaMA | 7B | 12.37 | 5.30 | 44.67 | P1264, P3403, P3095 | P1542, P926, P3275 |
| GLM | 10B | 5.59 | 5.12 | 24.05 | P2922, P2499, P397 | P3091, P681, P210 |
| Dolly | 12B | 15.60 | 11.35 | 39.88 | P1264, P3403, P2822 | P1542, P926, P3275 |
| LLaMA | 13B | 13.86 | 6.02 | 45.36 | P1264, P3403, P3095 | P1542, P926, P3275 |
| Alpaca | 13B | 8.24 | 3.77 | 31.5 | P770, P1906, P2974 | P1427, P674, P1951 |
| Vicuna | 13B | 19.50 | 8.34 | 49.83 | P410, P974, P2159 | P263, P1542, P1906 |
| WizardLM | 13B | 16.90 | 1.13 | 50.24 | P2839, P521, P3085 | P926, P1427, P922 |
| Moss | 16B | 11.20 | 12.83 | 27.01 | P1264, P3403, P1537 | P1542, P184, P926 |
| LLaMA | 65B | 14.56 | 6.98 | 44.80 | P3261, P1576, P2551 | P263, P1542, P1906 |
| LLaMA2 | 65B | 19.71 | 10.13 | 45.78 | P2839, P974, P3261 | P1542, P1906, P926 |
| OPT | 175B | 23.06 | 19.89 | 42.31 | P3403, P1264, P2922 | P2321, P2746, P538 |

Table 6: Detailed results of statistical knowledge assessment on 20 LLMs.

# D   Detailed Results

Tab. 6 provides detailed results for statistical knowledge assessment on LLMs.

# E   Case Study on LLM Generation

Tab. 7 provides a case study on LLM generation.

# F   Human Annotation Specification

In this section, we provide further details on our human annotation.

## F.1   Main Procedures

In our Human Annotation Assessment (HKA), there are two primary procedures: generation-guided prompting and knowledge mastery scoring. Each procedure is carried out by a group of three volunteer annotators who have backgrounds in NLP.

### F.1.1   Generation-Guided Prompting

In our HKA, a critical procedure is generation-guided prompting. That is, annotators continually modify their prompts according to both manual and automatic feedback from the model generations.

| Model | Question | Generation |
|---|---|---|
| LLaMA | The occupation of Swan of Avon was: | The Swan of Avon was a British cargo ship... |
| | William Shakespeare worked as a | playwright and actor in London |
| | Joseph Armstrong worked for: | The Daily Telegraph (London) |
| | The companies where Joseph Armstrong worked in are: | 1. The Armstrong Company, 1900-1901
2. The Armstrong Company, 1901-1902... |
| Alpaca | The occupation of Swan of Avon was: | that of a boatman... |
| | William Shakespeare worked as a | a playwright, poet, and actor... |
| | Joseph Armstrong worked for: | Joseph Armstrong is a former employee of the following companies:
Joseph Armstrong currently works for:... |
| | The companies where Joseph Armstrong worked in are: | Joseph Armstrong's average salary was $100,000... |
| Vicuna | The occupation of Swan of Avon was: | The Swan of Avon was a paddle steamer... |
| | William Shakespeare worked as a | a playwright and actor... |
| | Joseph Armstrong worked for: | Joseph Armstrong was a member of the following organisations:... |
| | The companies where Joseph Armstrong worked in are: | * Armstrong, Whitworth & Co.... |

Table 7: The generation cases of the original 7B LLaMA model and its instruction-tuned variants Alpaca and Vicuna. For each model, we provide the same Knowledge-related prompts.

This approach differs from previous scenarios where annotators merely wrote relation templates for each fact to create prompts [Petroni et al., 2019]. Instead, we ask each annotator to write prompts from various perspectives, including prompting for entities, relations, or the entire fact. Moreover, we perform manual answer-type checking throughout the annotation process. Adjusting prompts according to the model's generation is essential for accurately assessing the model's knowledge. For example, because of the prompt template "[X] is a [Y]" in LAMA, 100% of the top-1 prediction for facts of relation 'P31' is a definite article or a predicate. This induces the underestimation of model knowledge while implementing LAMA@1. And for 'P19', although the prompt in LAMA, "[X] was born in [Y]." is a plausible one, most of the top-10 generations are dates or months. This indicates the significance of model-generation-guided prompt refining.

**Manual Answer-type Checking**   To ensure that the prompt accurately prompts the model to generate the relevant text of the current fact in the next word, instead of freely describing other content, we conduct manual answer-type checking. Only when the model generations are mostly of the same type of target answers, the prompts are deemed valid ones; otherwise, they will be further refined to a valid one.

**Automatic Diversity Checking**   To explore the model generations towards different prompts, we implement an automatic diversity check for the prompts. Regarding the other valid prompts as references and the new prompt as the candidate, we calculate the BLEU score [Papineni et al., 2002] and set a similarity threshold to 0.8 to filter similar prompts. Besides, annotates are asked to not only write prompts to probe the object entity of each fact but also write prompts to probe the subject, the relation, or the whole fact.

### F.1.2   Knowledge Mastery Scoring

In the knowledge mastery scoring procedure, we enlist an additional three annotators to evaluate the model's knowledge. Provided with a prompt and the corresponding GPT2-XL model generation (top-1 generation using beam search), annotators are asked to score the model's knowledge mastery on a scale between 0 (unknown) and 1 (known). Ultimately, the scores for prompts related to the

same fact are averaged to determine the model's knowledge mastery score for that particular piece of factual knowledge.

## F.2  Annotation Specification

As it introduced above, our annotation mainly contains two parts: A. prompting, B. scoring. All the annotators are volunteer students majoring in Computational Linguistics in China. We told them the annotation intention and explained how the data would be used.

### F.2.1  Prompting

Given a triplet fact, annotators are instructed to do their best to determine whether the knowledge is stored in the LLM and compose three prompts (from various perspectives) for each fact to probe it.

During the prompting process, annotators should create prompts that not only probe the object entity but also examine the subject, the relation, or the entire fact. For instance, for the fact <Obama, birthplace, Hawaii>, possible prompts could include:

- *One U.S. president was born in Hawaii, and his name was ____ (This prompt probes for the subject)*
- *Obama was born in ____ (this prompt probes for the object).*
- *Hawaii is Obama's ____ (this prompt probes for the relation).*
- *The state where Barack Obama born is ____ (this prompt probes for the object).*
- *President Obama was born in Honolulu, ____ (this prompt probes for the object).*

To ensure the diversity of prompts, we employ the BLEU metric [Papineni et al., 2002] for automatic diversity checks. Specifically, we consider all valid prompts as references and the new prompt as the candidate, setting a threshold of 0.5 to filter out prompts that are similar to existing ones in terms of surface forms.

Annotators assess whether a given prompt effectively elicits knowledge based on the model-generated results. If the generated content is primarily related to the target aspect, the prompt is considered valid. For instance, when prompting for object entities, if the model's output consistently includes a set of entities (both correct and incorrect ones are acceptable, as long as the type is broadly relevant), the prompt is deemed valid. However, if the generated text or tokens consist of stopwords or unrelated entity types, the prompt is classified as invalid. For example, the fact <China, capital, Beijing>, *The capital city of China is ____* is not a valid prompt for GPT2 as the generations are: "inching", "home to", "is now a", "is described as" or "is the birthplace". While *The capital city of China is called ____* is a valid prompt as the model generations are city names like: "Shanghai", "Beijing" or "Hangzhou".

### F.2.2  Scoring

The scoring criteria are as follows:

- "Invalid": Factually unrelated generations or improper prompts.
- 1: The output produced during the greedy decoding process contains at least one alias of the correct answer, without any factually incorrect answers.
- 0: The generated content does not include any related entities or relations, nor are there any correct answers expressed in paraphrased words or phrases. In other words, the correct answer is not present in any of the generated responses.

## F.3  Annotation Results

We ultimately obtained 4,140 manually annotated prompts for 410 valid facts. The overall sentence similarity among the prompts for the same fact is 0.78, which demonstrates good diversity compared to the 0.95 overall similarity of prompts in ParaRel [Elazar et al., 2021]. The Kappa score between human raters is 0.4.

While HKA provides relatively trustworthy and reliable results for assessing model knowledge correctness, it places high demands on human annotators and is time-consuming to implement for

each fact in the model-specific annotation process. In our experience, annotating 100 facts on GPT2-XL takes an average of 4 hours per annotator. Consequently, HKA is more suitable as a solution for obtaining gold-standard results for knowledge assessment on a smaller scale. Kendall's $\tau$ correlation between the results of a knowledge-assessing approach and HKA reflects the evaluation's accuracy.

