# OpenReview forum: "Statistical Knowledge Assessment for Large Language Models"
_NeurIPS.cc/2023/Conference — NeurIPS 2023 poster_

### Official Review · Reviewer_4epC · 2023-07-05

**Soundness:** 3 good
**Presentation:** 3 good
**Contribution:** 2 fair
**Rating:** 6
**Confidence:** 3

**Summary:**

This paper proposes a statistical approach called KaRR to assess the factual knowledge contained in Generative Language Models (GLMs). The authors use a large-scale assessment suite with 994,123 entities and 600 relations to evaluate 14 GLMs of various sizes. The results show that KaRR exhibits a strong correlation with human assessment and achieves a lower variance to varying prompts. The experiments also reveal interesting insights into the scaling law of GLMs and the impact of tuning on instruction-following data.

**Strengths:**

1. The proposed statistical approach is novel and effective in assessing the factual knowledge contained in GLMs. The proposed method effectively considers different surface forms of subject, object, and relation.
2. The large-scale assessment suite used in the experiments is comprehensive and orders of magnitude larger than prior studies.1. Some of the paper's findings, such as scaling laws of knowledge and fine-tuning instructions to improve consistency, have been found in previous work.
2. Previous work has systematically assessed the consistency of the same facts under multiple prompts, but the facts considered in this paper are larger in scale and prompts are more diverse.

**Weaknesses:**

1. Some of the paper's findings, such as scaling laws of knowledge and fine-tuning instructions to improve consistency, have been found in previous work.
2. Previous work has systematically assessed the consistency of the same facts under multiple prompts, but the facts considered in this paper are larger in scale and prompts are more diverse.

**Questions:**

N/A

---

> ### Author Rebuttal · Authors · 2023-08-09
>
> We really appreciate your effort in reviewing our paper and your acknowledgment of our paper’s contribution. We are very glad that you liked our statistical approach, the large-scale assessment suite, and the insightful findings.
>
> Our response to your further comments is as follows:
>
> **Q1 "Some of the paper's findings, such as scaling laws of knowledge and fine-tuning instructions to improve consistency, have been found in previous work."**
>
> We want to clarify our motivation for scaling the model size and highlight that our results are different from the findings in previous work.
>
> **Scaling laws of knowledge:**
> There may be some misunderstanding. Our primary objective in examining the evaluation results of scaling model size is to demonstrate that the knowledge assessment findings align with previous studies on scaling laws, thereby validating the performance of our approach.
>
> In addition, our research offers some unique insights. Figure in Table 2(b) first shows the scaling law of model knowledge of the different curves of scaling laws of different large generative language models. Besides, Figure 5 illustrates the detailed differences in model knowledge across different relations, providing an interpretation of the disparities among models of 350M, 2.7B, and 175B.
>
> **Fine-tuning instructions:**
> Yes, previous studies have mentioned fine-tuning instructions to consistency but our finding is novel that tuning on instruction-following data may compromise the model's capability to generate factually correct text consistently. However, our novel finding suggests that tuning based on instruction-following data might actually impede the model's ability to consistently generate factually accurate text.
>
> While instruction-tuning does enhance the understanding of instructions, it is not surprising that it also improves the general consistency of LLMs in response to prompts, as evidenced by earlier research. Our study, conversely, examines the consistency of knowledge, specifically in generating factually accurate answers consistently. Contrary to general consistency, our results reveal potential trade-offs between instruction understanding ability and consistent mastery of knowledge. We further present intriguing examples of this point in Table 1, located in Appendix 4.
>
> **Other findings:**
> In addition to these two findings, we have also uncovered many other findings that have not been previously investigated. For example, the spurious correlation in knowledge assessment (Table 3 (b)), the knowledge evaluation variance towards different prompts (Table 3 (a)), etc. We hope that these findings will enhance the research community's understanding of the knowledge assessment of GLMs and encourage further investigation.
>
> Again, we thank you for acknowledging that our approach is novel and effective, and experimental results are insightful. If you have any additional suggestions or concerns, we would be happy to discuss them further with you.
>
> **Reference:**
>
> [1] Petroni, Fabio, et al. Language models as knowledge bases?
>
> [2] Dhingra, Bhuwan, et al. Time-aware language models as temporal knowledge bases.
>
> [3] Roberts, Adam, Colin Raffel, and Noam Shazeer. How much knowledge can you pack into the parameters of a language model?

---

> > ### Author Response · Authors · 2023-08-17
> > **Follow up to Reviewer 4epC**
> >
> > Dear Reviewer 4epC,
> >
> > We would like to thank you again for your reviews and your acknowledgment of our novelty. We have added replies to the weakness you mentioned and highlighted our other findings.
> >
> > Since the rebuttal deadline is approaching soon,  we would appreciate it if you could let us know if our responses have addressed your concerns satisfactorily. If your concerns have not been resolved, could you please let us know about it so that we have the opportunity to respond before the deadline?
> >
> > We would really appreciate it if you are willing to increase your score. And we would be happy to have any follow-up discussions or address any additional concerns.
> >
> > Thanks very much! Looking forward to your reply.
> >
> > Paper9365 Autors

---

### Official Review · Reviewer_zX87 · 2023-07-06

**Soundness:** 3 good
**Presentation:** 3 good
**Contribution:** 3 good
**Rating:** 7
**Confidence:** 4

**Summary:**

The paper introduces an automatic evaluation metric to assess the amount of factual knowledge kept by large language models (LLMs). This proposed metric considers various surface forms of factual knowledge presentation, allowing for an evaluation that not only measures the accuracy of the models in terms of factual knowledge but also considers the prediction robustness of the knowledge. Later, the authors demonstrated that this metric has a higher correlation with human annotations than previously proposed metrics. Additionally, the paper includes several robustness analyses for the metric, such as examining the impact of the prompting format and its relationship to co-occurrence statistics. And it shows the new metric's effectiveness compared to previous metrics.

**Strengths:**

1. The proposed metric is commendable as it goes beyond capturing mere accuracy and considers model performance consistency. This aspect is crucial in evaluating the factual knowledge of large language models, as it reflects their ability to provide correct information consistently.
2. The strong invariance results achieved by the automatic evaluation metric compared to other baselines are a notable strength.
3. The paper's comprehensive analysis of the proposed metric is great. By covering multiple aspects, the authors thoroughly evaluate the metric's performance. This level of analysis contributes to a better understanding of the metric's strengths, limitations, and overall effectiveness in assessing the factual knowledge of large language models.

**Weaknesses:**

1. The human correlation score of 0.43 for the automatic evaluation metric may be considered relatively low.
2. The proposed metric is not interpretable. To improve the interpretability of the score, calculating an oracle score and then comparing the current score to the oracle score ratio could be a useful approach. This ratio can aid in better understanding the effectiveness of the metric and provide a clearer picture of its performance relative to the best achievable outcome.

**Questions:**

1. In The KaRR scoring details section in Appendix, you mentioned the selection of a threshold based on human judgment alignment. It would be valuable to explore the impact of different threshold values on the metric's performance and investigate the generalizability of the chosen threshold on other sets of knowledge. Does certain threshold value also result in high correlation in other datasets?
2. Minor suggestion: compress Figure 5 by representing the average performance of the best and worst relations. This modification could improve the readability of the figure and make it easier to interpret the performance trends.
3. Adding the best and worst relations to the appendix would help understand the model performance. I would be curious to see it personally.
4. There is a missing number in the equation mentioned in line 43 of the appendix.

**Limitations:**

No, the limitation of the work is not discussed.

---

> ### Author Rebuttal · Authors · 2023-08-09
>
> We sincerely thank Reviewer zX87 for the positive recommendation as well as the valuable suggestions.
> We really appreciate your kind words that our metric is commendable and our analysis is comprehensive. Below we would like to give detailed responses to each of your comments.
>
> **Q1 "The human correlation score of 0.43 for the automatic evaluation metric may be considered relatively low."**
>
> We'd like to clarify that 0.43 Kendall's tau correlation is a relatively high score for an automatic metric. To resolve possible misunderstandings, it's worth mentioning Kendall's tau correlation scores in [1][2].
>
> As our evaluation of the KaRR metric is similar to the segment-level metric evaluation for machine translation, below we list Kendall's tau scores for some widely accepted metrics for machine translation evaluation for comparison. As shown, Kendall's tau is a relatively strict metric compared to Pearson’s correlation, and 0.43 is a relatively high score for automatic metrics.
>
> Metric  | Pearson’s correlation |  Kendall's tau |
> ------------- | ------------- |  ------------- |
> METEOR  | 0.484  | 0.324 |
> NLEPOR | 0.483  | 0.281  |
> SENTBLEU-MOSES | 0.465  | 0.266  |
> DEP-REF-EX  | 0.453  | 0.307   |
> |
>
> *(Pearson’s correlation and Kendall’s τ between WMT-13 segment-level metrics and human assessment for Spanish-to-English. Please refer to Table 2, Sec 4.2 in [1] for the whole table.)*
> Metric  | Kendall's tau |
> ------------- | ------------- |
> HTER  | 0.4324 |
>  HMEANT gold - monolinguals \* |0.4324  |
> HMEANT auto - monolinguals \* | 0.3964  |
> BLEU / METEOR / TER / PER | 0.1982  |
> |
>
> *(Sentence-level correlation with human adequacy judgments. The weights for individual roles in the metric are tuned by optimizing the correlation.  Please refer to Table 8, Sec. 8.2 in [2] for the whole table.)*
>
> In addition, we compared the Kendall-tau correlation of KaRR with humans and the baseline metrics for model knowledge evaluation such as LAMA with humans in Table 1 (b). KaRR shows a much stronger correlation (KaRR: 0.43  versus LAMA@1: 0.17, K-prompts: 0.32). Besides, the Recall of finding human-detected false knowledge in  Table 1 (b) (KaRR: 95.18% versus LAMA@1: 83.25%, K-prompts: 78.00%) further support KaRR‘s correlation with human evaluation.
>
> **Q2 "To improve the interpretability of the score, calculating an oracle score and then comparing the current score to the oracle score ratio could be a useful approach. "**
>
> Many thanks for your insightful suggestion!
> Yes, we agree that calculating an oracle score and then comparing the current score to the oracle score ratio would be more interpretable. However, it is worth noting that obtaining the oracle score for each GLM on every piece of knowledge is hardly feasible. We can only sample a portion of knowledge and obtain the human evaluation results (Sec. 4.4), but the cost of the human evaluation is substantial.
> To improve the interpretability of the score, we have updated our draft. In the latest version, we have included the quantitative results for both $KaRR_r$ and $KaRR_s$.
>
> **Q3 "In the KaRR scoring details section in Appendix, you mentioned the selection of a threshold based on human judgment alignment. It would be valuable to explore the impact of different threshold values on the metric's performance and investigate the generalizability of the chosen threshold on other sets of knowledge. Does certain threshold value also result in high correlation in other datasets?"**
>
> Thank you for making a great point. As suggested, we add the experiments on different threshold values and the generalizability of the chosen threshold.
>
> The results below highlight the significance of selecting a human-aligned threshold for evaluation accuracy, as minimal human input enhances the correlation with human assessments. If the threshold is too low, the criteria become lenient, and variance slightly increases. Conversely, an excessively high threshold results in near-100% Recall for false knowledge detection due to strictness but slightly reduces the correlation with human judgment.
>
> Encouragingly, our approach displays commendable generalization for the chosen threshold on other sets of knowledge. This implies that once we select a threshold in alignment with human judgment in a specific knowledge base, it can be directly applied to other knowledge bases.
>
>  Threshold  | KaRR Score | Variance | Recall | Kendall's \tau |
> ------------- | ------------- |  ------------- |------------- | ------------- |
> 22 (\*human-aligned)  | 12.27  | 0.67  | 95.18  | 0.43  |
> 8  | 33.05  | 0.87  |  84.20  | 0.28  |
> 16  | 17.20 | 0.69  | 90.05  | 0.36  |
> 32 |  7.93  | 0.64  | 99.53  | 0.32  |
> |
>
> *(Impact of different threshold values.)*
>
>  Threshold  | KaRR Score | Variance | Recall | Kendall's \tau |
> ------------- | ------------- |  ------------- | ------------- |  ------------- |
> T-REx  | 12.27  | 0.67  | 0.67  | 95.18 | 0.43  |
> Google-RE  |  9.76| 0.54  | 91.78  | 0.39   |
> ConceptNet | 10.02  | 0.68  | 87.06  | 0.39    |
> SQuAD  | 9.98  | 0.80  | 85.21  |  0.34  |
> |
>
> *(Generalizability of the chosen threshold on other sets of knowledge. For the three new databases, facts are randomly sampled and manually aligned since some of the knowledge bases do not correspond with entity ids in Wikidata.)*
>
> Many thanks for your constructive suggestions on Figure 5 and the appendix. As suggested, we've added the best and worst relations for each GLM in Table 2(a) to Appendix 6,  and added the equation number "(8)" in line 43 of the appendix.
>
> Thanks again for your detailed and constructive comments! We hope our answers have addressed your concerns.
>
> **Reference:**
>
> [1] Graham, Yvette, Timothy Baldwin, and Nitika Mathur. Accurate evaluation of segment-level machine translation metrics.
>
> [2] Lo, Chi-kiu, and Dekai Wu. MEANT: An inexpensive, high-accuracy, semi-automatic metric for evaluating translation utility based on semantic roles.

---

> > ### Comment · Reviewer_zX87 · 2023-08-15
> > **Thanks for the reply**
> >
> > I've read the author's response and will keep my score. I appreciate the explanation of Kendall's tau score and the additional analysis!

---

> > > ### Author Response · Authors · 2023-08-17
> > > **Thanks for the positive feedback**
> > >
> > > We sincerely thank you for the positive feedback and we are grateful for the time you spent on our submission and rebuttal. We are also delighted that the explanation of Kendall's tau score and the analysis have been acknowledged. We hope our paper can provide contributions to further understanding and exploring the knowledge of GLMs. Thanks again!

---

### Official Review · Reviewer_adWh · 2023-07-06

**Soundness:** 3 good
**Presentation:** 3 good
**Contribution:** 2 fair
**Rating:** 5
**Confidence:** 3

**Summary:**

The paper proposes KaRR, a statistical approach to assess factual knowledge for generative language models based on graphical models. An assessment suite is also proposed for future research. Experiments are conducted with 14 popular large language models and comprehensive analyses are also conducted to reveal related properties.

**Strengths:**

- This paper released a dataset for assessing factual knowledge in generative language models, which is large-scale (millions of entities and text aliases) and could be used in future works.
- This paper proposed a statistical score KaRR based on graphical models and KaRR aligns well with human preferences as shown with human evaluation.
- Lots of popular large language models are experimented with and the analyses are interesting, e.g., Table 2(b).

**Weaknesses:**

- The knowledge assessment focuses on entity-aware knowledge, which could be a relatively limited knowledge form. The current LLMs are good at identifying entity-aware knowledge. The major hallucinations actually come from numbers, dates, etc. Not a strict weakness, just wondering if KaRR could be extended to these aspects.
- There are in total 600 relations in the assessment suite. Does that mean the suite can only be employed for the specific 600 relations? What if there are new relations?

**Questions:**

-  The conclusion of "instruction tuning impairs knowledge reliability" should be considered more carefully (line 236). There are at least two factors contributing to the KaRR difference between Alpaca and Vicuna, data quality and tuning style. You do not know exactly whether data or tuning style is the major reason.
- Minor comments:
  - line 228, repeated "KaRR score".
  - line 230, should be "with a 3.92 KaRR score difference"
  - GLM in Table 2 and GLM across the paper are a little bit confusing.
- What are the distributions of relations in the dataset?


**Limitations:**

Refer to previous weaknesses and questions.

---

> ### Author Rebuttal · Authors · 2023-08-09
>
>
> We sincerely thank Reviewer adWh for the positive comments on our method and analyses as well as the valuable suggestions. We would like to give detailed responses to each of your comments.
>
> **Q1. “The knowledge assessment focuses on entity-aware knowledge, which could be a relatively limited knowledge form. The current LLMs are good at identifying entity-aware knowledge. The major hallucinations actually come from numbers, dates, etc. Not a strict weakness, just wondering if KaRR could be extended to these aspects.”**
>
> Thank you for this interesting point. Yes, we do rely on entity-aware knowledge; however, our definition of "entity" encompasses a wide range of categories, including dates, natural features, numbers, laws, and more. As a result, KaRR already incorporates the evaluation of facts associated with such diverse entities.
>
> In fact, dates can have aliases as well. For example, the fact <Barack Obama, date_of_birth, 4 August 1961 (Q69285218)> can be expressed in various ways within the prompt, including "August 4, 1961," "4 August 1961," and "1961-08-04." Similarly, for numbers, the number 7 (Q23350) can be represented as "the number 7," "seven," "number seven," "number 7," and so on.
>
> **Q2. “There are in total 600 relations in the assessment suite. Does that mean the suite can only be employed for the specific 600 relations? What if there are new relations?”**
>
> Thank you for highlighting the potential confusion. We would like to clarify that our statistical knowledge assessment approach is not limited to the 600 relations used in our experiments. The graphical model for knowledge assessment and the KaRR metric, as described in Section 3, can be implemented with various entities or relation types. For instance, researchers focusing on medical relations can easily substitute relations with predicates from a medical knowledge graph, such as RepoDB[1] and SemMedDB[2].
>
> If new relations emerge, we can generate relation templates for these new relations following the same process outlined in Section 4.1, and subsequently incorporate them into the relation and fact sets. As our method is designed to serve as a general framework for GLM knowledge assessment, it can be flexibly adapted to different relation types based on specific requirements.
>
>
> **Q3. “The conclusion of "instruction tuning impairs knowledge reliability" should be considered more carefully (line 236). There are at least two factors contributing to the KaRR difference between Alpaca and Vicuna, data quality and tuning style. You do not know exactly whether data or tuning style is the major reason.”**
>
> Thanks for the great suggestion and sorry for the confusion. The conclusion is derived from the comparison of **the original LLaMA and the Alpaca**, as the Alpaca is finetuned on the LLaMA with instruction-following data.
>
> We agree that there are other factors contributing to the KaRR difference between Alpaca and Vicuna and we have utilized the phrase "could lead to" (Line 234) to convey our hypothesis regarding the potential cause.
> To be more rigorous, we have modified the claims accordingly in the revised version (i.e., "For larger GLMs, a comparison between the original LLaMA and the Alpaca model reveals that instruction-tuning might influence the model's ability to generate consistent and correct knowledge" ).
>
> **Q4 "What are the distributions of relations in the dataset?"**
>
> Thanks for this great point! We've incorporated the following table into our Appendix 7. The distribution of relations exhibits a long-tail phenomenon, which is consistent with the distribution of relations in real-world knowledge and the distribution of relations within knowledge graphs[3].
>
> Top-k  | \# Avg. aliases| \# Related facts | Proportion of related facts |
> ------------- | ------------- |  ------------- | ------------- |
> 50  | 6.96  | 12638356   | 90.05\% |
> 100  | 5.97  | 13538967   | 96.47\% |
> 150  | 5.54  | 13802467  | 98.35\% |
> 200  | 5.22  | 13921742  | 99.20\% |
> 250  | 4.84  | 13977899   | 99.60\% |
> |
>
> *(Distributions of top-k relations in the dataset.)*
>
> Moreover, Figure 2 in the T-REx paper[1] demonstrates the distribution of the number of alignments created for each relation within the T-REx dataset. T-REx notably surpasses other datasets in terms of the number of examples provided, not only for the most prevalent predicates but also for those found in the long tail.
>
> **Q5. Typo and writing**
>
> Thank you for your gracious help in identifying typos and writing problems in our work. We have implemented all required changes in the newest version.
>
> **Reference:**
>
> [1] Brown, Adam S., and Chirag J. Patel. A standard database for drug repositioning.
>
> [2] Kilicoglu, Halil, et al. SemMedDB: a PubMed-scale repository of biomedical semantic predications
>
> [3] Elsahar, Hady, et al. T-REx: A large scale alignment of natural language with knowledge base triples

---

> > ### Author Response · Authors · 2023-08-17
> > **Further comments and discussions will be appreciated!**
> >
> > Dear Reviewer  adWh,
> >
> >
> > Thank you for your valuable time to review our work and for your constructive feedback. We posted our response to your comments a week ago, and we wonder if you could kindly share some of your thoughts so we can keep the discussion rolling to address your concern if there are any.
> >
> >
> > In the previous response,
> >
> > 1. We clarified the scope of our "entity", which encompasses a wide range of categories, including dates and numbers as you mentioned. This enables our knowledge assessment to cover a wide range of knowledge.
> >
> > 2. We outlined the procedure for employing our method when new relations emerge. It is important to note that our method can be flexibly adapted to different relation types based on specific requirements.
> >
> > 3. To eliminate any misunderstanding, we explained the reasoning behind the conclusion on instruction-tuning and, as suggested, revised the claim to be more rigorous.
> >
> > 4. We addressed the distribution of relations in the dataset by providing a table on the proportion of related facts for the top-k relations. This table has been incorporated into Appendix 7 in the revised version of our paper. And we've modified the typos and writing problems you mentioned, thank you for your gracious help again.
> >
> >
> > We would appreciate it if you could kindly take a look at both the revision and our response to your comments. We would really appreciate it if you are willing to increase your score. If you have any further questions, we are happy to discuss them!
> >
> > Best regards,
> >
> > Authors

---

> > ### Comment · Reviewer_adWh · 2023-08-18
> > **Thanks for the response.**
> >
> > I have read the author's responses and other reviewers' responses as well. I appreciate the further statistics and explanations for my questions. It would be great to include them in a future version of this paper.

---

### Official Review · Reviewer_e9uD · 2023-07-07

**Soundness:** 3 good
**Presentation:** 2 fair
**Contribution:** 2 fair
**Rating:** 4
**Confidence:** 3

**Summary:**

this paper proposes a statistical method to probe the knowledge in generative language models, which aims at connecting symbolic knowledge and GLM's text format generation. More specifically, the KaRR comprises two components with regard to specifying relation and subject entity. The authors also present the graphical model for model implementation on the text. The results show that the knowledge in GLMs follows the scaling law, but when the model is finetuned on instruction-following data, it may compromise the model's ability to consistently generate factually correct text.

**Strengths:**

1. very important and interesting problem, assessing the knowledge stored in generative language models is challenging and worth studying
2. the proposed KaRR method shows strong robustness to prompt variance
3. Interesting findings: instruction-following data compromises the model’s capability to generate factually correct answer.

**Weaknesses:**

1. This work only focuses on generating the object, I believe the author can assess the model's ability to generate the subject (by reversing the triples/facts) to gain a more comprehensive assessment of knowledge stored in GLMs.
2. The average alias count for each subject is approx 1.39, and for each object it is about 1.78, which I believe could not support the "diverse" claim.
3. The proposed KaRR’s design to measure reliability is not clear enough.
4. In line 38, the authors claim "Prior methods are designed for masked language models (MLMs) and are incapable of measuring GLMs.". However, to the best of my knowledge, there are several works about assessing the knowledge in GLMs, such as [1, 2].

[1] Hendrycks, D., Burns, C., Basart, S., Zou, A., Mazeika, M., Song, D., & Steinhardt, J. (2020). Measuring massive multitask language understanding. arXiv preprint arXiv:2009.03300.
[2] Dhingra, B., Cole, J. R., Eisenschlos, J. M., Gillick, D., Eisenstein, J., & Cohen, W. W. (2022). Time-aware language models as temporal knowledge bases. Transactions of the Association for Computational Linguistics, 10, 257-273.


**Questions:**

1. I believe it would be interesting to see the model's knowledge across different domains, such as encyclopedia, biomedical, etc.
2. Since high-quality aliases are often difficult to obtain, it would be interesting to see the impact of alias, such as alias count, etc.

---

> ### Author Rebuttal · Authors · 2023-08-09
>
> We sincerely thank Reviewer e9uD for your review and are grateful for the time you spent on our submission. We are also glad you think our research problem is important and our findings are interesting. Below we would like to give detailed responses to each of your comments.
>
> **Q1 “This work only focuses on generating the object, I believe the author can assess the model's ability to generate the subject (by reversing the triples/facts) to gain a more comprehensive assessment of knowledge stored in GLMs.”**
>
> Thank you for your comments. It is worth mentioning that the reversed facts have already been included  in our knowledge base, T-REx. For example, both <Barack Obama (Q76), spouse, Michelle Obama (Q13133)> and <Michelle Obama (Q13133), spouse, Barack Obama (Q76)> are covered. Please note that not all facts can be reversed, so the final number of subject and object entities we obtain are slightly different.
>
> Moreover, our approach is intrinsically not restricted to particular entity types, as long as corresponding triplets are constructed. Formally, our method incorporates the probabilities $P(e_2|e_1, r)$, $P(e_2|r)$, and $P(e_2|e_1)$, where $e_1$ or $e_2$ can serve as either subject or object and maintain symmetric properties. We implement this method by employing naturally existing triplets within the current knowledge base, which covers a wide range of knowledge.
>
> **Q2 "The average alias count for each subject is approx 1.39, and for each object it is about 1.78, which I believe could not support the "diverse" claim.”**
>
> Thank you for highlighting the potential confusion. We'd like to clarify that real-world alias distribution exhibits a long-tailed pattern, with a substantial portion of entities having only one alias. This significantly affects the average number of aliases.
>
> However, the most frequent entities typically possess more than five aliases, related to a large proportion of facts (details shown in the table below).
>
> |\# Top-k  | Type | \# Avg. aliases | Proportion of related facts |
> |------------- | ------------- |  ------------- | ------------- |
> |50  | subject   | 7.62  | 12.10\% |
> |100  | subject    | 6.73  | 16.13\% |
> |500 | subject   | 4.68  | 28.34\% |
> |1000  |subject   | 3.70  | 33.70\% |
> |50  | object  | 5.06 | 39.84\% |
> |100  | object  | 4.79  |  47.35\% |
> |500 | object  | 3.80  | 65.44\% |
> |1000  | object | 3.48 | 71.93\% |
> |
>
> **Q3 “KaRR’s design to measure reliability is not clear enough.”**
>
> As mentioned in Lines 24-29 and illustrated in Fig. 1, reliability refers to the ability to consistently generate knowledge-correct text towards various possible prompts with the same semantics.
>
> To measure reliability, we take multiple text forms for the same entity or relation into knowledge assessment and build the graphical model of text forms and symbolic knowledge (triplets). As mentioned in Sec. 3.3, we expand the KaRR metrics based on the graphical model (Eq. 4-9), so as to evaluate GLM on a simple knowledge with various possible prompts.
>
> **Q4 “There are several works about assessing the knowledge in GLMs, such as [1, 2].”**
>
> Thank you for mentioning related works. We'd like to clarify that existing methods, both open-form and closed-form, are tailored for specific models or tasks, but not well-suited for a comprehensive knowledge assessment of most GLMs.
>
> Open-form methods like TEMPLAMA [1] probe factual knowledge in MLMs and an exceptional GLM, T5, using a cloze-test format. However, they're not applicable to most GLMs without masked token/span prediction objectives (see Lines 35-42).
>
> Closed-form methods, such as multiple-choice questions [2], evaluate domain-specific knowledge-utilization and problem-solving abilities rather than assessing knowledge itself. These methods may also be biased toward specific option numbers.
>
> To avoid confusion, we've revised the statement to: "Prior methods target MLMs and don't provide a universal solution for assessing GLMs' knowledge."
>
> **Q5 "I believe it would be interesting to see the model's knowledge across different domains, such as encyclopedia, biomedical, etc."**
>
> Thank you for your constructive comments. We agree that it is valuable to assess and analyze model knowledge across different domains. Due to the lack of domain-specific entity aliases, we focus on encyclopedia knowledge using T-REx and Wikidata.
>
> Nonetheless, our proposed graphical model and KaRR metric provide versatile solutions for knowledge assessment, adaptable to diverse domains by replacing the knowledge base with relevant domain-specific data. This flexible approach sets the stage for future research in domain-specific knowledge evaluation.
>
> **Q6 "Since high-quality aliases are often difficult to obtain, it would be interesting to see the impact of alias, such as alias count, etc."**
>
> Thank you for making a great point. As suggested, we add the experiments using different numbers of aliases. The results (listed in the following table) show that a larger number of aliases decrease the variance of knowledge assessment, which is consistent with our intuition and the analysis of the sampling number  K in Sec. 7.
>
> \# Avg. aliases  | KaRR Score | Variance |
> ------------- | ------------- |  ------------- |
> 1  | 19.67  | 0.96   |
> 2  | 18.98  | 0.72  |
> 4  | 18.78  | 0.55  |
> 8  | 18.82  | 0.51  |
> |
>
> *(GPT2-XL on 500\*20 facts of 500 frequent entities.)*
>
> Overall, we greatly appreciate your efforts for your thoughtful comments on our paper. We hope our answers have addressed your concerns. We have revised the paper to clear the issues you mentioned in the comments in our latest version.
>
> **Reference:**
>
> [1] Hendrycks, D., Burns, C., Basart, S., Zou, A., Mazeika, M., Song, D., & Steinhardt, J. (2020). Measuring massive multitask language understanding.
>
> [2] Dhingra, B., Cole, J. R., Eisenschlos, J. M., Gillick, D., Eisenstein, J., & Cohen, W. W. (2022). Time-aware language models as temporal knowledge bases.

---

### Official Review · Reviewer_bA4R · 2023-07-14

**Soundness:** 4 excellent
**Presentation:** 4 excellent
**Contribution:** 3 good
**Rating:** 7
**Confidence:** 3

**Summary:**

The paper presents a framework for the quantitative assessment of the knowledge captured by large language models, which includes a proposed metric and a large set of relations. Given subject-relation-object triplets, a basic approach to quantify the model's knowledge would be to use the probability of the object entity being generated, given the subject and the relation. This naive approach has shortcomings, which the authors address in the following way:
1. Entities can have multiple synonyms, which are important to consider to accurately assess consistency. The authors therefore augment their proposed evaluation suite with a set of aliases for each entity, extracted from Wikidata, and adapt the metric to take these into account.
2. As this kind of probing often suffers from spurious correlations, the authors propose the "knowledge assessment risk ratio" metric, which  also takes into account the expected generation probability when either the subject or the relation entity are not specified.

The authors thoroughly evaluate the proposed approach on 14 generative model. They additionally measure the effectiveness of their approach compared to other pre-existing metrics, showing strong correlation with human judgement and robustness of the metric towards prompt variation.

**Strengths:**

* This is a sound and extensively evaluated approach towards the automatic quantitative assessment of the knowledge learned by LLMs. This method displays strong correlation with human judgements. While it cannot replace human annotation entirely, it should be extremely useful to the community as a cheaper, faster automatic way of performing knowledge assessment, much like how BLEU can be used during model development to validate the performance of translation models.
* The authors should be commended for calling out robustness (consistency of generation given similar prompts). This aspect is crucial for real-world applications of such models, and is often ignored in similar works.

**Weaknesses:**

* The proposed framework is limited to assessing knowledge in a fairly simplistic way, via the prediction of entities in subject-relation-object triplets. It would have been interesting to hear more about the limitations of such an approach, taking into account e.g. slightly more complex type of queries (see e.g. arXiv:2305.01157).

Very minor:
* lines 141-144, 148, 202, 291, 313-315: straight quotes should be turned into curly quotes

**Questions:**

* Do you think your proposed approach could be effective as a validation metric for the training of LLMs?

**Limitations:**

The authors did a decent job of addressing limitations, and I don't expect any potential negative societal impact from this work.

---

> ### Author Rebuttal · Authors · 2023-08-09
>
> We sincerely thank Reviewer bA4R for the positive feedback and we are grateful for the time you spent on our submission. We are also glad for the acknowledgment that the problem we are working on is realistic and that the method we propose is sound. We would like to provide comprehensive responses to your comments and questions.
>
> **Q1 "The proposed framework is limited to assessing knowledge in a fairly simplistic way, via the prediction of entities in subject-relation-object triplets. It would have been interesting to hear more about the limitations of such an approach, taking into account e.g. slightly more complex types of queries (see e.g. arXiv:2305.01157)."**
>
> Many thanks for your constructive comments! Yes, our knowledge assessment focuses on atomic knowledge--each piece of knowledge consists of a triplet.
>
> As suggested, we outline several limitations of the current method as follows:
> 1. struggle to evaluate the knowledge that requires multi-hop reasoning or the understanding of complex relationships between multiple entities.
> 2.  struggle to evaluate model knowledge mastery with context-dependent or time-varying information, which is often crucial for correct knowledge understanding and representation.
>
> Thanks for your advice again! We've incorporated them into our limitation section, and we're interested in eliminating the limitations in future work.
>
> **Q2 "Do you think your proposed approach could be effective as a validation metric for the training of LLMs?"**
>
> Thank you for making a great point. Our approach can be well-suited as a knowledge evaluation metric during LLM training, as our testing approach aligns with the model's next token prediction objective. It can be tested across various base models, such as the LLaMA, GPT2-XL, and T5-large reported in Table 2.
>
> It's noteworthy that the goals of the language model training phase go beyond just model knowledge - they might also include semantic understanding, among others. These goals and the learning of knowledge may involve certain trade-offs. If we directly apply KaRR as a validation metric, it could lead to new issues. Nevertheless, through the implementation of reasonable improvements to our metrics, we can overcome these issues.
>
> We appreciate your constructive feedback and are keen to explore this perspective further.
>
> **Q3. Typo and format.**
>
> Thanks for kindly pointing out our typos and format problems. We have revised them all in the latest version.
>
> Thank you very much for the constructive comments, which really help us further improve our work. We hope our answers have addressed your concerns. If you have any further questions, we are happy to address them.

---

> > ### Author Response · Authors · 2023-08-17
> > **Further comments and discussions will be appreciated!**
> >
> > Dear Reviewer bA4R,
> >
> > We would like to thank you again for your detailed and constructive reviews. To better address your concerns regarding "the limitations of such an approach," we have incorporated discussions on two additional limitations in the latest version of our draft.
> >
> > Furthermore, we appreciate your reference to the related work on more complex query types. We have included a discussion on the paper mentioned[1] in both our Related Work and Limitation sections, as well as a discussion on the possible extensions of our approach for such queries (i.e., complex query decomposition and a compound metric for complex knowledge) in our future work discussion.
> >
> > For other questions, we have updated our draft and added replies to your comments. Overall, many thanks for your insightful points and suggestions. These comments really help improve our paper. We hope our answers have addressed your concerns. If you have any further questions, we are happy to address them. We would really appreciate it if you are willing to increase your score.
> >
> > Thanks very much!
> >
> > Best regards,
> >
> > Authors
> >
> > **Reference:**
> >
> > [1] Choudhary, Nurendra, and Chandan K. Reddy. "Complex Logical Reasoning over Knowledge Graphs using Large Language Models." arXiv preprint arXiv:2305.01157 (2023).

---

> > ### Comment · Reviewer_bA4R · 2023-08-21
> > **Response to rebuttal**
> >
> > Many thanks to the authors for the responses to my questions. Having read the other reviews and rebuttals, I confirm my scores.

---

### Decision · Program_Chairs · 2023-09-21

**Decision:**

Accept (poster)

**Comment:**

This paper presents a statistical approach to determine whether generative language models (GLMs) can produce factual knowledge. It concentrates on examining knowledge expressed as subject-verb-object triplets, and also provides a dataset for testing such knowledge. The paper also provides an extensive evaluation of recent models, as well as insights into how instruction tuning affects the performance of models in terms of producing factual knowledge. This evaluation and the proposed dataset can be used to inform future research on the factuality of GLMs. I encourage the authors to update the paper to clarify the points discussed with the reviewers, and to also elaborate on the generality of the proposed method, as it focuses on evaluating knowledge that can be expressed in a triplet format.